# A Mechanistic Interpretation of Arithmetic Reasoning in Language Models using Causal Mediation Analysis

**Alessandro Stolfo**
ETH Zürich
stolfoa@ethz.ch

**Yonatan Belinkov**
Technion – IIT, Israel
belinkov@technion.ac.il

**Mrinmaya Sachan**
ETH Zürich
msachan@ethz.ch

## Abstract

Mathematical reasoning in large language models (LMs) has garnered significant attention in recent work, but there is a limited understanding of how these models process and store information related to arithmetic tasks within their architecture. In order to improve our understanding of this aspect of language models, we present a mechanistic interpretation of Transformer-based LMs on arithmetic questions using a causal mediation analysis framework. By intervening on the activations of specific model components and measuring the resulting changes in predicted probabilities, we identify the subset of parameters responsible for specific predictions. This provides insights into how information related to arithmetic is processed by LMs. Our experimental results indicate that LMs process the input by transmitting the information relevant to the query from mid-sequence early layers to the final token using the attention mechanism. Then, this information is processed by a set of MLP modules, which generate result-related information that is incorporated into the residual stream. To assess the specificity of the observed activation dynamics, we compare the effects of different model components on arithmetic queries with other tasks, including number retrieval from prompts and factual knowledge questions.[1]

## 1 Introduction

Mathematical reasoning with Transformer-based models (Vaswani et al., 2017) is challenging as it requires an understanding of the quantities and the mathematical concepts involved. While large language models (LMs) have recently achieved impressive performance on a set of math-based tasks (Wei et al., 2022a; Chowdhery et al., 2022; OpenAI, 2023), their behavior has been shown to be inconsistent and context-dependent (Bubeck et al., 2023). Recent literature shows a multitude

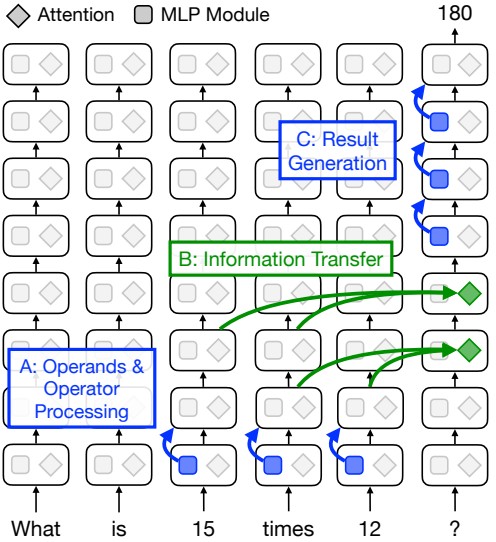

Figure 1: Visualization of our findings. We trace the flow of numerical information within Transformer-based LMs: given an input query, the model processes the representations of numbers and operators with early layers (A). Then, the relevant information is conveyed by the attention mechanism to the end of the input sequence (B). Here, it is processed by late MLP modules, which output result-related information into the residual stream (C).

of works proposing methods to improve the performance of large LMs on math benchmark datasets through enhanced pre-training (Spokoyny et al., 2022; Lewkowycz et al., 2022; Liu and Low, 2023) or specific prompting techniques (Wei et al., 2022b; Kojima et al., 2022; Yang et al., 2023, *inter alia*). However, there is a limited understanding of the inner workings of these models and how they store and process information to correctly perform math-based tasks. Insights into the mechanics behind LMs' reasoning are key to improvements such as inference-time correction of the model's behavior (Li et al., 2023) and safer deployment. Therefore, research in this direction is critical for the development of more faithful and accurate next-generation LM-based reasoning systems.

In this paper, we present a set of analyses aimed

---

[1] Our code and data is available at https://github.com/alestolfo/lm-arithmetic.

at mechanistically interpreting LMs on the task of answering simple arithmetic questions (e.g., *"What is the product of 11 and 17?"*). In particular, we hypothesize that the computations involved in reasoning about such arithmetic problems are carried out by a specific subset of the network. Then, we test this hypothesis by adopting a causal mediation analysis framework (Vig et al., 2020; Meng et al., 2022), where the model is seen as a causal graph going from inputs to outputs, and the model components (e.g., neurons or layers) are seen as mediators (Pearl, 2001). Within this framework, we assess the impact of a mediator on the observed output behavior by conducting controlled interventions on the activations of specific subsets of the model and examining the resulting changes in the probabilities assigned to different numerical predictions.

Through this experimental procedure, we track the flow of information within the model and identify the model components that encode information about the result of arithmetic queries. Our findings show that the model processes the input by conveying information about the operator and the operands from mid-sequence early layers to the final token using attention. At this location, the information is processed by a set of MLP modules, which output result-related information into the residual stream (shown in Figure 1). We verify this finding for bi- and tri-variate arithmetic queries across four pre-trained language models with different sizes: 2.8B, 6B, and 7B parameters. Finally, we compare the effect of different model components on answering arithmetic questions to two additional tasks: a synthetic task that involves retrieving a number from the prompt and answering questions related to factual knowledge. This comparison validates the *specificity* of the activation dynamics observed on arithmetic queries.

## 2 Related Work

**Mechanistic Interpretability.** The objective of mechanistic interpretability is to reverse engineer model computation into components, aiming to discover, comprehend, and validate the algorithms (called circuits in certain works) implemented by the model weights (Räuker et al., 2023). Early work in this area analyzed the activation values of single neurons when generating text using LSTMs (Karpathy et al., 2015). A multitude of studies have later focused on interpreting weights and intermediate representations in neural networks (Olah

et al., 2017, 2018, 2020; Voss et al., 2021; Goh et al., 2021) and on how information is processed by Transformer-based (Vaswani et al., 2017) language models (Geva et al., 2021, 2022, 2023; Olsson et al., 2022; Nanda et al., 2023). Although not strictly mechanistic, other recent studies have analyzed the hidden representations and behavior of inner components of large LMs (Belrose et al., 2023; Gurnee et al., 2023; Bills et al., 2023).

**Causality-based Interpretability.** Causal mediation analysis is an important tool that is used to effectively attribute the causal effect of mediators on an outcome variable (Pearl, 2001). This paradigm was applied to investigate LMs by Vig et al. (2020), who proposed a framework based on causal mediation analysis to investigate gender bias. Variants of this approach were later applied to mechanistically interpret the inner workings of pre-trained LMs on other tasks such as subject-verb agreement (Finlayson et al., 2021), natural language inference (Geiger et al., 2021), indirect object identification (Wang et al., 2022), and to study their retention of factual knowledge (Meng et al., 2022).

**Math and Arithmetic Reasoning.** A growing body of work has proposed methods to analyze the performance and robustness of large LMs on tasks involving mathematical reasoning (Pal and Baral, 2021; Piękos et al., 2021; Razeghi et al., 2022; Cobbe et al., 2021; Mishra et al., 2022). In this area, Stolfo et al. (2023) use a causally-grounded approach to quantify the robustness of large LMs. However, the proposed formulation is limited to behavioral investigation with no insights into the models' inner mechanisms. To the best of our knowledge, our study represents the first attempt to connect the area of mechanistic interpretability to the investigation of the mathematical reasoning abilities in Transformer-based LMs.

## 3 Methodology

### 3.1 Background and Task

We denote an autoregressive language model as $\mathcal{G} : \mathcal{X} \rightarrow \mathcal{P}$. The model operates over a vocabulary $V$ and takes a token sequence $x = [x_1, ..., x_T] \in \mathcal{X}$, where each $x_i \in V$. $\mathcal{G}$ generates a probability distribution $\mathbb{P} \in \mathcal{P} : \mathbb{R}^{|V|} \rightarrow [0, 1]$ that predicts possible next tokens following the sequence $x$. In this work, we study decoder-only Transformer-based models (Vaswani et al., 2017). Specifically, we focus on models that represent a slight variation of

the standard GPT paradigm, as they utilize parallel attention (Wang and Komatsuzaki, 2021) and rotary positional encodings (Su et al., 2022). The internal computation of the model's hidden states $h_t^{(l)}$ at position $t \in \{1, \ldots, T\}$ of the input sequence is carried out as follows:

$$h_t^{(l)} = h_t^{(l-1)} + a_t^{(l)} + m_t^{(l)} \qquad (1)$$
$$a_t^{(l)} = \mathrm{A}^{(l)}\left(h_1^{(l-1)}, \ldots, h_t^{(l-1)}\right)$$
$$m_t^{(l)} = W_{\mathrm{proj}}^{(l)} \, \sigma\left(W_{fc}^{(l)} \, h_t^{(l-1)}\right)$$
$$=: \mathrm{MLP}^{(l)}(h_t^{(l-1)}),$$

where at layer $l$, $\sigma$ is the sigmoid nonlinearity, $W_{fc}^{(l)}$ and $W_{\mathrm{proj}}^{(l)}$ are two matrices that parameterize the multilayer perceptron (MLP) of the Transformer block and $\mathrm{A}^{(l)}$ is the attention mechanism.[2]

We consider the task of computing the result of arithmetic operations. Each arithmetic query consists of a list of operands $N = (n_1, n_2, \ldots)$ and a function $f_O$ representing the application of a set of arithmetic operators $(+, -, \times, \div)$. We denote as $r = f_O(N)$ the result obtained by applying the operators to the operands. Each query is rendered as a natural language question through a prompt $p(N, f_O) \in \mathcal{X}$ such as *"How much is $n_1$ plus $n_2$?"* (in this case, $f_O(n_1, n_2) = n_1 + n_2$). The prompt is then fed to the language model to produce a probability distribution $\mathbb{P}$ over $V$. Our aim is to investigate whether certain hidden state variables are more important than others during the process of computing the result $r$.

### 3.2 Experimental Procedure

We see the model $\mathcal{G}$ as a causal graph (Pearl, 2009), framing internal model components, such as specific neurons, as mediators positioned along the causal path connecting model inputs and outputs. Following a causal mediation analysis procedure, we then quantify the contribution of particular model components by intervening on their activation values and measuring the change in the model's output. Previous work has isolated the effect of every single neuron within a model (Vig et al., 2020; Finlayson et al., 2021). However, this approach becomes impractical for models with billions of parameters. Therefore, for our main experiments, the elements that we consider as variables along the causal path described by the model are

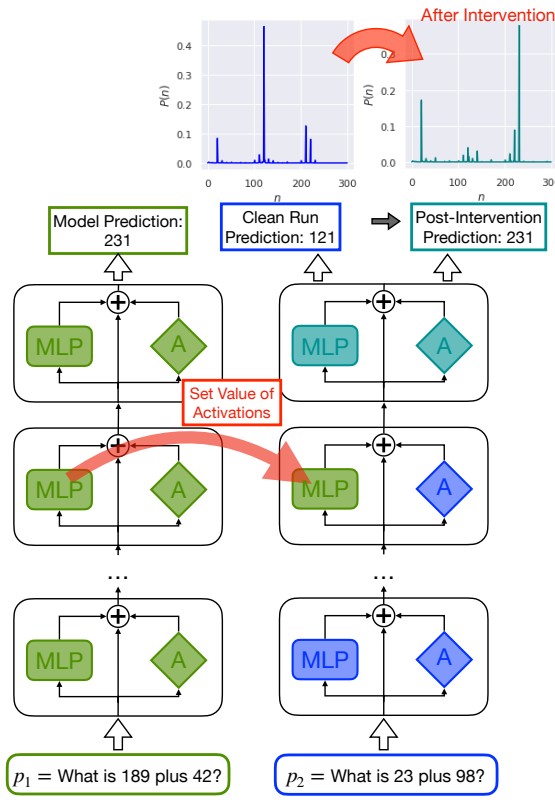

Figure 2: By intervening on the activation values of specific components within a language model and computing the corresponding effects, we identify the subset of parameters responsible for specific predictions.

the outputs of the $\mathrm{MLP}^{(l)}$ and $\mathrm{A}^{(l)}$ functions at each token $t$, i.e., $m_t^{(l)}$ and $a_t^{(l)}$.

To quantify the importance of modules $\mathrm{MLP}^{(l)}$ and $\mathrm{A}^{(l)}$ in mediating the model's predictions at position $t$, we use the following procedure.

1. Given $f_O$, we sample two sets of operands $N, N'$, and we obtain $r = f_O(N)$ and $r' = f_O(N')$. Then, two input questions with only the operands differing, $p_1 = p(N, f_O)$ and $p_2 = p(N', f_O)$, are passed through the model.

2. During the forward pass with input $p_1$, we store the activation values $\bar{m}_t^{(l)} := \mathrm{MLP}^{(l)}(h_t^{(l-1)})$, and $\bar{a}_t^{(l)} := \mathrm{A}^{(l)}(h_1^{(l-1)}, \ldots, h_t^{(l-1)})$ .

3. We perform an additional forward pass using $p_2$, but this time we *intervene* on components $\mathrm{MLP}^{(l)}$ and $\mathrm{A}^{(l)}$ at position $t$, setting their activation values to $\bar{m}_t^{(l)}$, and $\bar{a}_t^{(l)}$, respectively. This process is illustrated in Figure 2.

4. We measure the causal effect of the intervention on variables $m_t^{(l)}$ and $a_t^{(l)}$ on the model's prediction by computing the change in the probability values assigned to the results $r$ and $r'$.

---

[2]For brevity, layer normalization (Ba et al., 2016) is omitted as it is not essential for our analysis.

More specifically, we compute the **indirect effect** (IE) of a specific mediating component by quantifying its contribution in skewing $\mathbb{P}$ towards the correct result. Consider a generic activation variable $z \in \{m_1^{(1)}, \ldots, m_t^{(L)}, a_1^{(1)}, \ldots, a_t^{(L)}\}$. We denote the model's output probability following an intervention on $z$ as $\mathbb{P}_z^*$. Then, we compute the IE as:

$$\text{IE}(z) = \frac{1}{2}\left[\frac{\mathbb{P}_z^*(r) - \mathbb{P}(r)}{\mathbb{P}(r)} + \frac{\mathbb{P}(r') - \mathbb{P}_z^*(r')}{\mathbb{P}_z^*(r')}\right] \quad (2)$$

where the two terms in the sum represent the relative change in the probability assigned by the model to $r$ and $r'$, caused by the intervention performed. The larger the measured IE, the larger the contribution of component $z$ in shifting probability mass from the clean-run result $r'$ to result $r$ corresponding to the alternative input $p_1$.[3]

We additionally measure the mediation effect of each component with respect to the operation $f_O$. We achieve this by fixing the operands and changing the operator across the two input prompts. More formally, in step 1, we sample a list of operands $N$ and two operators $f_O$ and $f_O'$. Then, we generate two prompts $p_1 = p(N, f_O)$ and $p_2 = p(N, f_O')$ (e.g., "*What is the sum of 11 and 7?*" and "*What is the product of 11 and 7?*"). Finally, we carry out the procedure in steps 2–4.

### 3.3 Experimental Setup

We present the results of our analyses in the main paper for GPT-J (Wang and Komatsuzaki, 2021), a 6B-parameter pre-trained LM (Gao et al., 2020). Additionally, we validate our findings on Pythia 2.8B (Biderman et al., 2023), LLaMA 7B (Touvron et al., 2023), and Goat, a version of LLaMA fine-tuned on arithmetic tasks (Liu and Low, 2023). We report the detailed results for these models in Appendix J.

In our experiments, we focus on two- and three-operand arithmetic problems. Similar to previous work (Razeghi et al., 2022; Karpas et al., 2022), for single-operator two-operand queries, we use a set of six diverse templates representing a question involving each of the four arithmetic operators. For the three-operand queries, we use one template for each of the 29 possible two-operator combinations. Details about the templates are reported in

Appendix A. In the bi-variate case, for each of the four operators $f_O \in \{+, -, \times, \div\}$ and for each of the templates, we generate 50 pairs of prompts by sampling two pairs of operands $(n_1, n_2) \in \mathcal{S}^2$ and $(n_1', n_2') \in \mathcal{S}^2$, where $\mathcal{S} \subset V \cap \mathbb{N}$. For the operand-related experiment, we sample $(n_1, n_2)$ and a second operation $f_O'$. In both cases, we ensure that the result $r$ falls within $\mathcal{S}$.[4] In the three-operand case, we generate 15 pairs of prompts for each of the 29 templates, following the same procedure. In order to ensure that the model achieves a meaningful task performance, we use a two-shot prompt in which we include two exemplars of question-answer for the same operation that is being queried. We report the accuracy results in Appendix B.

## 4 Causal Effects on Arithmetic Queries

Our analyses address the following question:

**Q1** What are the components of the model that mediate predictions involving arithmetic computations?

We address this question by first studying the flow of information throughout the model by measuring the effect of each component (MLP and attention block) at each point of the input sequence for two-operand queries (§4.1). Then, we distinguish between model components that carry information about the result and about the operands of the arithmetic computations (§4.2 and §4.3). Finally, we consider queries involving three operands (§4.4) and present a measure to quantify the changes in information flow (§4.5).

### 4.1 Tracing the Information Flow

We measure the indirect effect of each MLP and attention block at different positions along the input sequence. The output of these modules can be seen as new information being incorporated into the residual stream. This new information can be produced at any point of the sequence and then conveyed to the end of the sequence for the prediction of the next token. By studying the IE at different locations within the model, we can identify the modules that generate new information relevant to the model's prediction. The results are reported in Figures 3a and 3b for MLP and attention, respectively.

---

[3]As an alternative metric to quantify the IE, we experiment using the difference in log probabilities (Appendix I). The results obtained with the two metrics show consistency and lead to the same conclusions.

[4]Unless otherwise specified, we use $\mathcal{S} = \{1, 2, \ldots, 300\}$, as larger integers get split into multiple tokens by the tokenizer.

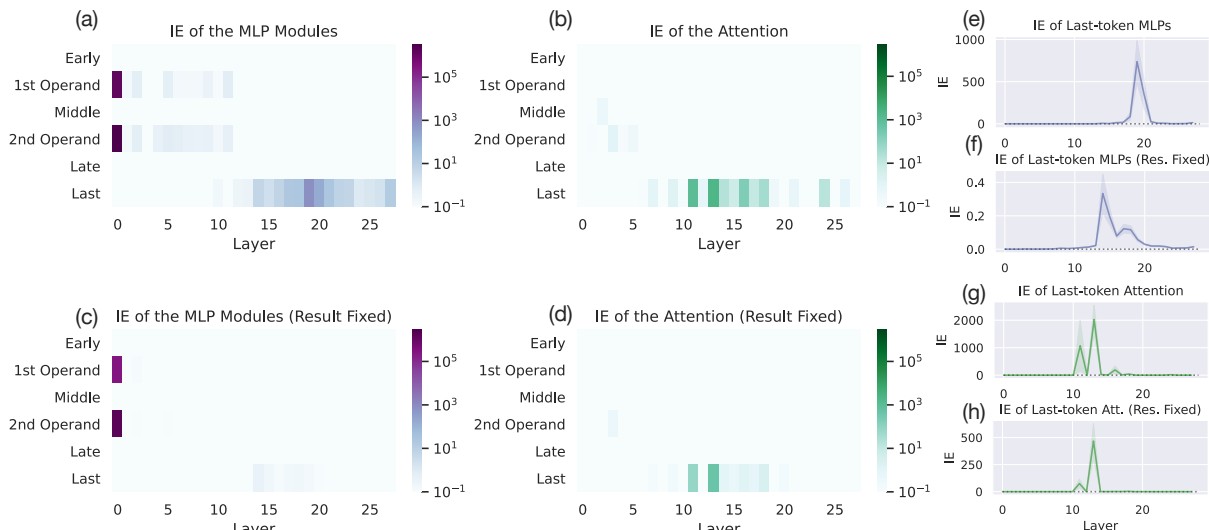

Figure 3: Indirect effect (IE) measured within GPT-J. Figures (a) and (b) illustrate the flow of information related to both the operands and the result of the queries, while the effect displayed in Figures (c) and (d) is related to the operands only (the result is kept unchanged). Figures (e–h) show a re-scaled visualization of the effects at the last token for each of the four heatmaps (a–d). The difference in the effect registered for the MLPs at layers 15–25 between figures (a) and (c) illustrates the role of these components in producing result-related information.

Our analysis reveals four primary activation sites: the MLP module situated at the first layer corresponding to the tokens of the two operands; the intermediate attention blocks at the last token of the sequence; and the MLP modules in the middle-to-late layers, also located at the last token of the sequence. It is expected to observe a high effect for the first MLPs associated with the tokens that vary (i.e., the operands), as such modules are likely to affect the representation of the tokens, which is subsequently used for the next token prediction. On the other hand, of particular interest is the high effect detected at the attention modules in layers 11–18 and in the MLPs around layer 20.

As for the flow of information tied to the operator, the activations display a parallel pattern: high effect is registered at early MLPs associated with the operator tokens and at the same last-token MLP and attention locations. We report the visualization of the operator-related results in Appendix C.

A possible explanation of the model's behavior on this task is that the attention mechanism facilitates the propagation of operand- and operator-related information from the first layers early in the sequence to the last token. Here, this information is processed by the MLP modules, which incorporate the information about the result of the computation in the residual stream. This hypothesis aligns with the existing theory that attributes the responsibility of moving and extracting infor-

mation within Transformer-based models to the attention mechanism (Elhage et al., 2021; Geva et al., 2023), while the feed-forward layers are associated with performing computations, retrieving facts and information (Geva et al., 2022; Din et al., 2023; Meng et al., 2022). We test the validity of this hypothesis in the following section.

## 4.2 Operand- and Result-related Effects

Our objective is to verify whether the contribution to the model's prediction of each component measured in Figures 3a and 3b is due to (*1*) the component representing information related to the operands, or (*2*) the component encoding information about the result of the computation. To this end, we formulate a variant of our previous experimental procedure. In particular, we condition the sampling of the second pair of operands $(n'_1, n'_2)$ on the constraint $r = r'$. That is, we generate the two input questions $p_1$ and $p_2$, such that their result is the same (e.g., "*What is the sum of 25 and 7?*" and "*What is the sum of 14 and 18?*"). In case number (*1*), we would expect a component to have high IE both in the result-varying setting and when $r = r'$, as the operands are modified in both scenarios. In case (*2*), we expect a subset of the model to have a large effect when the operands are sampled without constraints but a low effect for the fixed-result setting.

We report the results in Figure 3c and 3d. By

comparing Figures 3a and 3c, two notable observations emerge. First, the high effect in the early layers corresponding to the operand tokens is observed in both the result-preserving and the result-varying scenarios. Second, the last-token mid-late MLPs that lead to a high effect on the model's prediction following a result change, dramatically decrease their effect on the model's output in the result-preserving setting, as described in scenario (2). These observations point to the conclusion that the MLP blocks around layer 20 incorporate result-relevant information. As for the contribution of the attention mechanism (Figures 3b and 3d), we do not observe a substantial difference in the layers with the highest IE between the two settings, which aligns this scenario to the description of case (1). These results are consistent with our hypothesis that operand-related information is transferred by the attention mechanism to the end of the sequence and then processed by the MLPs to obtain the result of the computation.

### 4.3 Zooming in on the Last Token

In Figures 3e–3h, we show a re-scaled version of the IE measurements for the layers at the end of the input sequence. While the large difference in magnitude was already evident in the previously considered visualizations, in Figures 3e and 3f we notice that the MLPs with the highest effect in the two settings differ: the main contribution to the model's output when the results are not fixed is given by layers 19 and 20, while in the result-preserving setting the effect is largest at layers 14-18. For the attention (Figures 3g and 3h), we do not observe a significant change in the shape of the curve describing the IE across different layers, with layer 13 producing the largest contribution. We interpret this as additional evidence indicating that the last-token MLPs at layers 19-20 encode information about $r$, while the attention modules carry information related to the operands.

### 4.4 Three-operand Queries & Fine-tuning

We extend our analyses by including three-operand arithmetic queries such as "*What is the difference between $n_1$ and the ratio between $n_2$ and $n_3$?*". Answering correctly this type of questions represents a challenging task for pre-trained language models, and we observe poor accuracy (below 10%) with GPT-J. Thus, we opt for fine-tuning the model on a small set of three-operand queries. The model that we consider for this analysis is Pythia 2.8B,

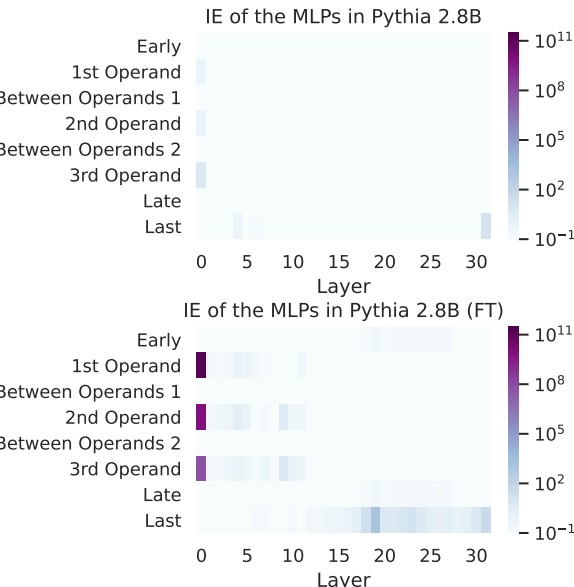

Figure 4: Indirect effect (IE) on three-operand queries for different MLP modules in Pythia 2.8B before and after fine-tuning. The effect produced by the last-token mid-late MLP activation site emerges with fine-tuning. Results for the attention are reported in Appendix J.

as its smaller size allows for less computationally demanding training than the 6B-parameter GPT-J. After fine-tuning, the model attains an accuracy of $\sim$40%. We provide the details about the training procedure in Appendix F.

We carry out the experimental procedure as in Section 4.1. In particular, we compare the information flow in the MLPs of the model before and after fine-tuning (Figure 4). In the non-fine-tuned version of the model, the only relevant activation site, besides the early layers at the operand tokens, is the very last layer at the last token. In the fine-tuned model, on the other hand, we notice the emergence of the mid-late MLP activation site that was previously observed in the two-operand setting.

### 4.5 Quantifying the Change of the Information Flow

Denote the set of MLPs in the model by $\mathcal{M}$. We define the relative importance (RI) of a specific subset $\mathcal{M}^* \subseteq \mathcal{M}$ of MLP modules as

$$\text{RI}(\mathcal{M}^*) = \frac{\sum_{m \in \mathcal{M}^*} \log(\text{IE}(m) + 1)}{\sum_{m \in \mathcal{M}} \log(\text{IE}(m) + 1)}. \quad (3)$$

In order to quantitatively show the difference in the activation sites observed in Figure 3, we compute the RI measure for the set

$$\mathcal{M}_{-1}^{\text{late}} = \{m_{-1}^{(\lfloor L/2 \rfloor)}, m_{-1}^{(\lfloor L/2 \rfloor + 1)}, \ldots, m_{-1}^{(L)}\},$$

| $|N|$ | Model | RI($\mathcal{M}_{-1}^{\text{late}}$) | RI($\mathcal{M}_{-1}^{\text{late}}$) Result Fixed |
|---|---|---|---|
| 2 | GPT-J | 40.2% | 4.4% |
| | Pythia 2.8B | 43.2% | 5.8% |
| | LLaMA 7B | 36.1% | 7.5% |
| | Goat | 33.5% | 7.4% |
| | GPT-J (Words) | 27.8% | 4.5% |
| 3 | Pythia 2.8B | 13.5% | 6.7% |
| | Pythia 2.8B (FT) | 24.7% | 13.6% |

Table 1: Relative importance (RI) measurements for the last-token late MLP activation site. The decrease in the RI observed when fixing the result of the two pairs of operands used for the interventions quantitatively confirms the role of this subset of the model in incorporating result-related information.

where the subscript $-1$ indicates the last token of the input sequence and $L$ is the number of layers in the model. This quantity represents the relative contribution of the mid-late last-token MLPs compared to all the MLP blocks in the model.

For the two-operand setting, we carry out the experimental procedure described in Section 3 for three additional models: Pythia 2.8B, LLaMA 7B, and Goat.[5] Furthermore, we repeat the analyses on GPT-J using a different number representation: instead of Arabic numbers (e.g., the token *2*), we represent quantities using numeral words (e.g., the token *two*). For the three-operand setting, we report the results for Pythia 2.8B before and after fine-tuning. We measure the effects using both randomly sampled and result-preserving operand pairs, comparing the RI measure in the two settings. The results (Table 1) exhibit consistency across all these four additional experiments. These quantitative measurements further highlight the influence of last-token late MLP modules on the prediction of $r$. We provide in Appendix J the heatmap illustrations of the effects for these additional studies.

## 5 Causal Effects on Different Tasks

In order to understand whether the patterns in the effect of the model components that we observed so far are specific to arithmetic queries, we compare our observations on arithmetic queries to two different tasks: the retrieval of a number from the prompt (§5.1), and the prediction of factual knowl-

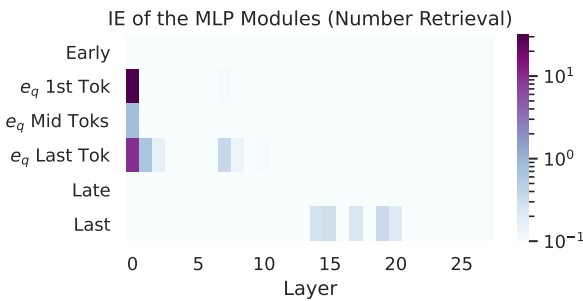

Figure 5: Indirect effect measured on the MLPs of GPT-J for predictions on the number retrieval task.

edge (§5.2). With this additional set of experiments, we aim to answer the question:

**Q2** Are the activation patterns observed so far *specific* to the arithmetic setting?

### 5.1 Information Flow on Number Retrieval

We consider a simple synthetic task involving numerical predictions. We construct a set of templates of the form "*Paul has $n_1$ $e_1$ and $n_2$ $e_2$. How many $e_q$ does Paul have?*", where $n_1$, $n_2$ are two randomly sampled numbers, $e_1$ and $e_2$ are two entity names sampled at random,[6] and $e_q \in \{e_1, e_2\}$. In this case, the two input prompts $p_1$ and $p_2$ differ solely in the value of $e_q$. To provide the correct answer to a query, the model has simply to *retrieve* the correct number from the prompt. With this task, we aim to analyze the model's behavior in a setting involving numerical predictions but not requiring any kind of arithmetic computation.

We report the indirect effect measured for the MLPs modules of GPT-J in Figure 5. In this setting, we observe an unsurprising high-effect activation site corresponding to the tokens of the entity $e_q$ and a lower-effect site at the end of the input in layers 14–20. The latter site appears in the set of the model components that were shown to be active on arithmetic queries. However, computing the relative importance of the late MLPs on this task shows that this second activation site is responsible for only RI($\mathcal{M}_{-1}^{\text{late}}$) = 8.7% of the overall $\log$ IE. The low RI, compared to the higher values measured on arithmetic queries, suggests that the function of the last-token late MLPs is not dictated by the numerical type of prediction, but rather by their involvement in *processing* the input information. This finding is aligned with our theory that

[5]The LLaMA tokenizer considers each digit as an independent token in the vocabulary. This makes it problematic to compare the probability value assigned by the model to multi-digit numbers. Therefore, we restrict the set of possible results to the set of single-digit numbers.

[6]We sample entities from a list containing names of animals, fruits, office tools, and other everyday items and objects.

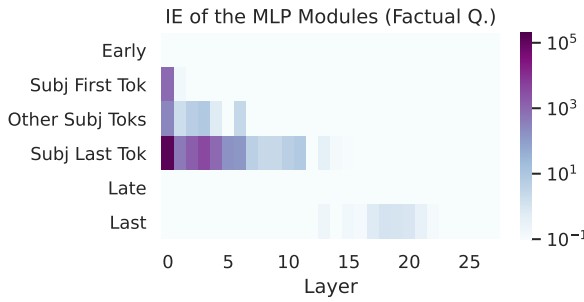

Figure 6: Indirect effect measured on the MLPs of GPT-J for predictions to factual queries.

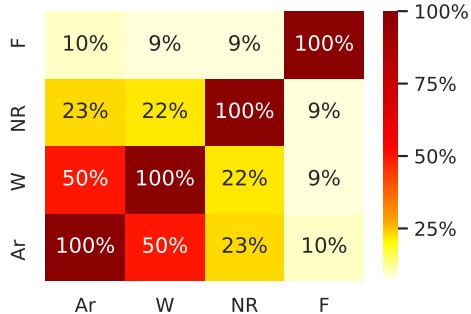

Figure 7: Overlap ratio in the top 400 neurons with the largest effect on predicting answers to factual queries involving Arabic Numerals (Ar) and numeral words (W), number retrieval (NR), and factual knowledge (F). The results are obtained with GPT-J.

sees $\mathcal{M}_{-1}^{\text{late}}$ as the location where information about $r$ is included in the residual stream.

## 5.2 Information Flow on Factual Predictions

We carry out our experimental procedure using data from the LAMA benchmark (Petroni et al., 2019), which consists of natural language templates representing knowledge-base relations, such as *"[subject] is the capital of [object]"*. By instantiating a template with a specific subject (e.g., *"Paris"*), we prompt the model to predict the correct object (*"France"*). Similar to our approach with arithmetic questions, we create pairs of factual queries that differ solely in the subject. In particular, we sample pairs of entities from the set of entities compatible for a given relation (e.g., cities for the relation "is the capital of"). Details about the data used for this procedure are provided in Appendix H. We then measure the indirect effect following the formulation in Equation 2, where the correct object corresponds to the correct numerical outcome in the arithmetic scenario.

From the results (Figure 6), we notice that a main activation site emerges in early layers at the tokens corresponding to the subject of the query. These findings are consistent with previous works (Meng et al., 2022; Geva et al., 2023), which hypothesize that language models store and retrieve factual associations in early MLPs located at the subject tokens. We compute the RI metric for the late MLP modules, which quantitatively validates the contribution of the early MLP activation site by attaining a low value of $\text{RI}(\mathcal{M}_{-1}^{\text{late}}) = 4.2\%$. The large IE observed at mid-sequence early MLPs represents a difference in the information flow with respect to the arithmetic scenario, where the modules with the highest influence on the model's prediction are located at the end of the sequence. This difference serves as additional evidence highlighting the

specificity of the model's activation patterns when answering arithmetic queries.

## 5.3 Neuron-level Interventions

The experimental results in Sections 5.1 and 5.2 showed a quantitative difference in the contributions of last-token mid-late MLPs between arithmetic queries and two tasks that do not involve arithmetic computation. Now, we investigate whether the components active *within* $\mathcal{M}_{-1}^{\text{late}}$ on the different types of tasks are different. We carry out a finer-grained analysis in which we consider independently each neuron in an MLP module (i.e., each dimension in the output vector of the function $\text{MLP}^{(l)}$) at a specific layer $l$. In particular, following the same procedure as for layer-level experiments, we intervene on each neuron by setting its activation to the value it would take if the input query contained different operands (or a different entity). We then compute the corresponding indirect effect as in Eq. 2. We carry out this procedure for arithmetic queries using Arabic numerals (Ar) and numeral words (W), for the number retrieval task (NR), and for factual knowledge queries (F).[7] We rank the neurons according to the average effect measured for each of these four settings and compute the overlap in the top 400 neurons (roughly 10%, as GPT-J has a hidden dimension of 4096).

We carry out this procedure for layer $l = 19$, as it exhibits the largest IE within $\mathcal{M}_{-1}^{\text{late}}$ on all the tasks considered. The heatmap in Figure 7 illustrates the results. We observe a consistent overlap (50%) between the top neurons active for the arithmetic queries using Arabic and word-based representa-

---

[7]To have the same result space for all the arithmetic queries (Ar and NW) and for the number retrieval task, we restrict the set $\mathcal{S}$ to $\{1, \ldots, 20\}$ (or the corresponding numeral words).

tions. Interestingly, the size of the neuron overlap between arithmetic queries and number retrieval is considerably lower (22% and 23%), even though both tasks involve the prediction of numerical quantities. Finally, the overlaps between the top neurons for the arithmetic operations and the factual predictions (between 9% and 10%) are not larger than for random rankings: the expected overlap ratio between the top 400 indices in two random rankings of size 4096 is 9.8% (Antverg and Belinkov, 2022). These results support the hypothesis that the model's circuits responsible for different kinds of prediction, though possibly relying on similar subsets of layers, are distinct. However, it is important to note that this measurement does not take into account the magnitude of the effect.

## 6 Conclusion

We proposed the use of causal mediation analysis to mechanistically investigate how LMs process information related to arithmetic. Through controlled interventions on specific subsets of the model, we assessed the impact of these mediators on the model's predictions.

We posited that models produce predictions to arithmetic queries by conveying the math-relevant information from the mid-sequence early layers to the last token, where this information is then processed by late MLP modules. We carried out a causality-grounded experimental procedure on four different Transformer-based LMs, and we provided empirical evidence supporting our hypothesis. Furthermore, we showed that the information flow we observed in our experiments is specific to arithmetic queries, compared to two other tasks that do not involve arithmetic computation.

Our findings suggest potential avenues for research into model pruning and more targeted training/fine-tuning by concentrating on specific model components associated with certain queries or computations. Moreover, our results offer insights that may guide further studies into using LMs' hidden representations to correct the model's behavior on math-based tasks at inference time (Li et al., 2023) and to estimate the probability of the model's predictions to be true (Burns et al., 2023).

## Limitations

The scope of our work is investigating arithmetic reasoning and we experiment with the four fundamental arithmetic operators. Addition, subtraction,

multiplication, and division form the cornerstone of arithmetic calculations and serve as the basis for a wide range of mathematical computations. Thus, exploring their mechanisms in language models provides a starting point to explore more complex forms of mathematical processing. Studying a broader set of mathematical operators represents an interesting avenue for further investigation.

Our work focuses on synthetically-generated queries that are derived from natural language descriptions of the four basic arithmetic operators. To broaden the scope, future research can expand the analysis of model activations to encompass math-based queries described in real-life settings, such as math word problems.

Finally, a limitation of our work concerns the analysis of different attention heads. In our experiments, we consider the output of an attention module as a whole. Future research could focus on identifying the specific heads that are responsible for forwarding particular types of information in order to offer a more detailed understanding of their individual contributions.

## Acknowledgments

AS is supported by armasuisse Science and Technology through a CYD Doctoral Fellowship. YB is supported by an AI Alignment grant from Open Philanthropy, the Israel Science Foundation (grant No. 448/20), and an Azrieli Foundation Early Career Faculty Fellowship. MS acknowledges support from the Swiss National Science Foundation (Project No. 197155), a Responsible AI grant by the Haslerstiftung, and an ETH Grant (ETH-19 21-1). We are grateful to Vilém Zouhar and Neel Nanda for the insightful discussions.

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

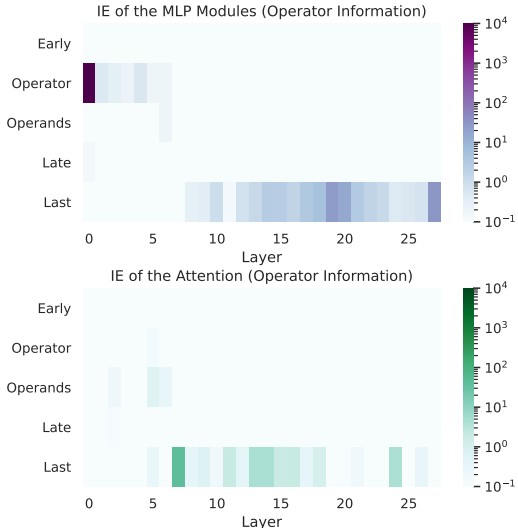

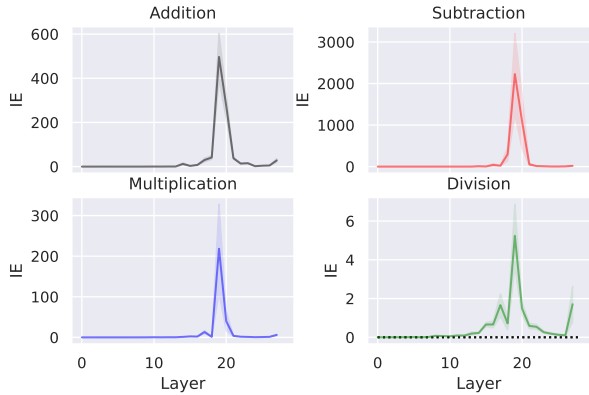

Figure 9: Indirect effect of the MLPs at the last token in each layer in GPT-J, for each of the four arithmetic operators. We observe a peak in the effect at layer 19 for all four types of operation.

Figure 8: Indirect effect (IE) measured in GPT-J when varying the word describing the operator involved in the input query. Similar to the operands case, we observe a high contribution produced by middle-to-late MLP modules at the end of the input sequence.

## A  Prompt Templates

In Tables 2 and 3, we report the question templates from Karpas et al. (2022), which we used as prompts for the model for two- and three-operand queries, respectively. For three-operand queries, we use one query template for each of the 29 possible two-operation combinations.

## B  Performance of the Models

In Table 4, we report the accuracy of the models on the arithmetic queries that we use for our analyses. The higher accuracy obtained using numeral words is likely given by the smaller set of possible solutions considered (we used $\mathcal{S} = \{$ *"one"*, *"two"*, ..., *"twenty"*$\}$, as the numeral words corresponding to larger numbers get split into multiple tokens by the tokenizer). The accuracy of GPT-J on the factual queries from the LAMA benchmark is 65.0% (we constrain the vocabulary to the set of all possible objects for all the relations considered). On the synthetic number retrieval task, GPT-J's accuracy is 86.7%.

## C  Flow of Operator-related Information

The measurements of the indirect effect for each model component when fixing the operand and varying the operator in the two input prompts $p_1$ and $p_2$ reveal how the model processes the information related to the operator. We report in Figure 8

the heatmap visualizations of these results for two-operand queries. Similar to the operand-related information, we observe a high effect in three activation locations: early MLP blocks corresponding to the operand tokens; middle-to-early attention modules at the last token; and middle-to-late MLP modules at the last token. These results align with our hypothesis that arithmetic-related information is transferred to the end of the sequence by the attention mechanism, where it is then processed by late MLP layers. In this setting, we measure $\text{RI}(\mathcal{M}_{-1}^{\textbf{late}}) = 31.4\%$.

## D  Effects for Each Operator

For each of the four operators, we report the indirect effect measured for the last-token MLP modules in GPT-J in Figure 9. The results for each of the four operators show a common spike in the effect at layers 19-20. This indicates the presence of a specific part of the model relevant to the numerical predictions of the bi-variate arithmetic questions, irrespective of the operator involved. We also notice a difference in the magnitude of the effects, which is linked to the capability of the model to correctly answer the query.

## E  Changes in the Model's Prediction

We measured the influence of the model components in terms of probability changes. Now, we study the dynamics of the actual model predictions. In particular, considering the scenario in which $r = r'$, we verify whether the intervention leads to a change in the model's prediction. That is, we

| Type | addition | subtraction |
|---|---|---|
| 1 | Q: How much is $n_1$ plus $n_2$? A: | Q: How much is $n_1$ minus $n_2$? A: |
| 2 | Q: What is $n_1$ plus $n_2$? A: | Q: What is $n_1$ minus $n_2$? A: |
| 3 | Q: What is the result of $n_1$ plus $n_2$? A: | Q: What is the result of $n_1$ minus $n_2$? |
| 3 | Q: What is the sum of $n_1$ and $n_2$? A: | Q: What is the difference between A: $n_1$ and $n_2$? A: |
| 5 | The sum of $n_1$ and $n_2$ is | The difference between $n_1$ and $n_2$ is |
| 6 | $n_1 + n_2 =$ | $n_1$ - $n_2 =$ |
| | **multiplication** | **division** |
| 1 | Q: How much is $n_1$ times $n_2$? A: | Q: How much is $n_1$ over $n_2$? A: |
| 2 | Q: What is $n_1$ times $n_2$? A: | Q: What is $n_1$ over $n_2$? A: |
| 3 | Q: What is the result of $n_1$ times $n_2$? A: | Q: What is the result of $n_1$ over $n_2$? A: |
| 4 | Q: What is the product of $n_1$ and $n_2$? A: | Q: What is the ratio between $n_1$ and $n_2$? A: |
| 5 | The product of $n_1$ and $n_2$ is | The ratio of $n_1$ and $n_2$ is |
| 6 | $n_1$ * $n_2 =$ | $n_1$ / $n_2 =$ |

Table 2: Question templates for two-operand arithmetic queries.

| Formula | Template |
|---|---|
| $(n_1+n_2)*n_3$ | Sum $n_1$ and $n_2$ and multiply by $n_3$ |
| $n_1+n_2*n_3$ | What is the sum of $n_1$ and the product of $n_2$ and $n_3$? |
| $(n_1-n_2)*n_3$ | What is the product of $n_1$ minus $n_2$ and $n_3$? |
| $n_1/(n_2/n_3)$ | How much is $n_1$ divided by the ratio between $n_2$ and $n_3$? |
| $n_1-n_2*n_3$ | What is the difference between $n_1$ and the product of $n_2$ and $n_3$? |
| $n_1*(n_2-n_3)$ | How much is $n_1$ times the difference between $n_2$ and $n_3$? |

Table 3: Examples of templates of three-operand queries. For the full list, we refer to Karpas et al. (2022).

| Model | Operation | Accuracy (%) |
|---|---|---|
| GPT-J | + | 69.3 |
| | − | 78.0 |
| | × | 82.8 |
| | ÷ | 40.8 |
| | Overall | 67.8 |
| GPT-J (Numeral Words) | + | 95.5 |
| | − | 86.7 |
| | × | 83.3 |
| | ÷ | 59.7 |
| | Overall | 81.3 |
| Pythia 2.8B | + | 57.4 |
| | − | 77.5 |
| | × | 64.7 |
| | ÷ | 40.2 |
| | Overall | 59.9 |
| LLaMA | + | 100.0 |
| | − | 99.8 |
| | × | 100.0 |
| | ÷ | 88.7 |
| | Overall | 97.2 |
| Goat | + | 100.0 |
| | − | 100.0 |
| | × | 91.4 |
| | ÷ | 54.0 |
| | Overall | 85.6 |
| Pythia 2.8B (3 Operands) | Overall | 0.9 |
| Pythia 2.8B Fine-tuned (3 Operands) | Overall | 39.7 |

Table 4: Accuracy of the models analyzed in the paper on various types of arithmetic queries.

compute

$$\mathbb{1}\{\arg\max_{x\in\mathcal{S}}\mathbb{P}_z^*(x) \neq \arg\max_{x\in\mathcal{S}}\mathbb{P}(x)\}, \quad (4)$$

distinguishing between desired ($\arg\max_{x\in\mathcal{S}}\mathbb{P}_*(x) = r$) and undesired ($\arg\max_{x\in\mathcal{S}}\mathbb{P}(x) = r$) changes. The results reported in Figure 10 show an increase in the desired change in prediction at layers 19-20, while the undesired change in prediction is higher for layers 14-17. This means that interventions on the MLPs at layers 19-20 are more likely to lead to a correct adjustment of the prediction, while the opposite is true for earlier layers (14-15 in particular). This finding is consistent with our previous observations and we see this as additional evidence highlighting the influence of the MLPs at layers 19-20 on the prediction of $r$.

## F  Fine-tuning Details

We fine-tune Pythia 2.8B on three-operand queries. We train the model for 2 epochs on a set of queries obtained by sampling 1000 triples of operands for each of the 29 templates. We use Adafactor (Shazeer and Stern, 2018) a learning rate of $10^{-5}$, linearly decaying, and a batch size of 8. We make sure that there is no overlap between the set of

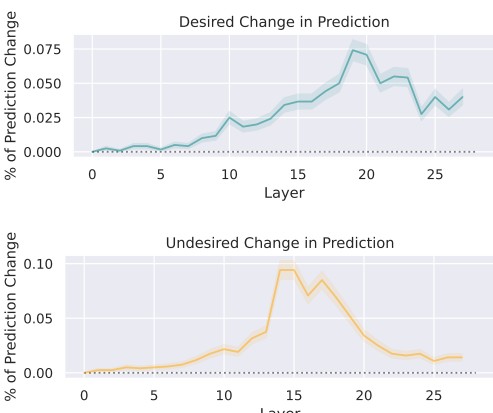

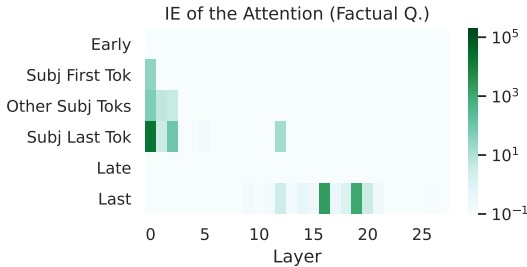

Figure 12: Indirect effect (IE) measured for the attention modules in GPT-J on factual knowledge queries.

## H  Factual Knowledge Data

For the experiments involving the prediction of factual knowledge, we use the following six relations from the T-REx subset of the LAMA benchmark (Petroni et al., 2019):

1. "[subject] is the capital of [object]"

2. "[subject] was born in [object]"

3. "[subject] died in [object]"

4. "The native language of [subject] is [object]"

5. "[subject] is a subclass of [object]"

6. "The capital of [subject] is [object]".

We sample pairs of subject entities from the set of entities compatible for a given relation (e.g., cities for the relation "is the capital of"). For each relation, we sample 100 pairs of subject entities.

## I  Log Probability to Quantify the IE

In order to validate whether the measurements of the indirect effect are specific to the metric that we describe in Equation 2, we quantify the IE using the absolute difference in the log of the probability values assigned by the model to the results $r$ and $r'$. More formally, we compute

$$ \text{IE}_{\text{alt}}(z) = \begin{cases} \Delta' + \Delta & \text{if } r \neq r' \\ |\Delta| & \text{otherwise} \end{cases}, \quad (5) $$

where

$$ \Delta' = \log \mathbb{P}^*_z(r) - \log \mathbb{P}(r) \quad (6) $$
$$ \Delta = \log \mathbb{P}(r') - \log \mathbb{P}^*_z(r'). \quad (7) $$

The results are reported in Figure 13. The activation sites that we observe are the same as reported in Section 4.1: first-layer MLP at the operand tokens and last-token MLP and attention modules.

Figure 10: Desired (wrong to correct) and undesired (correct to wrong) change in the prediction induced by the intervention on the MLPs in GPT-J. The layers at which the two types of prediction change peak correspond to the layers with the largest corresponding IE.

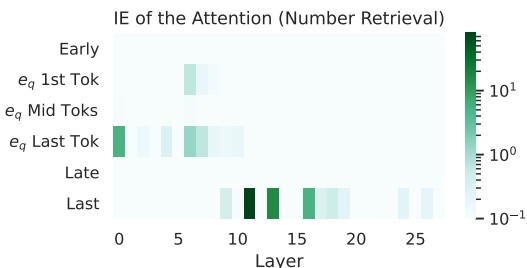

Figure 11: Indirect effect (IE) measured for the attention modules in GPT-J on the task of number retrieval.

queries used for training and the set used for the computation of the indirect effect.

## G  Computing Infrastructure

The experiments for all models are carried out using a single Nvidia A100 GPU with 80GB of memory. The computation of the indirect effect across the whole model for a single type of component (attention or MLP) took ∼15 hours for GPT-J and ∼6 hours for Pythia (using 50 examples for each two-operand template) and ∼7 hours for LLaMA and Goat (using 20 examples for each two-operand template). For the fine-tuning of Pythia 2.8B, we used a single Nvidia A100 GPU with 80GB of memory. The training procedure took ∼1 hour. Experiment tracking was carried out using Weights & Biases.[8]

---

[8] http://wandb.ai

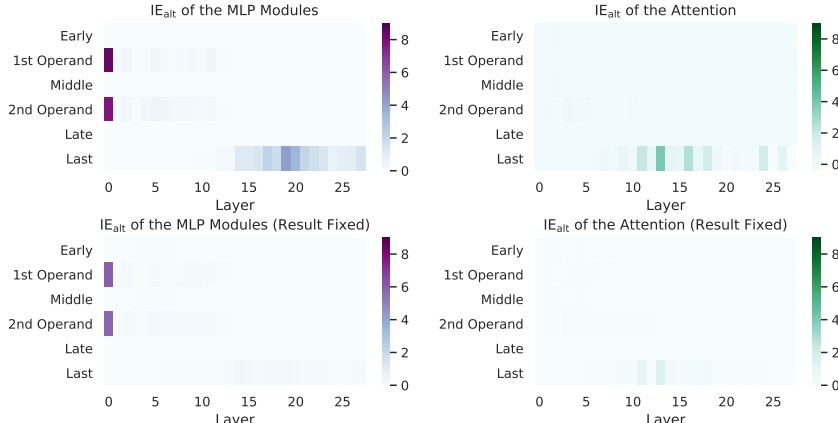

Figure 13: Indirect effect measured using the difference in the log probability as described in Equation 5 ($IE_{alt}$). The results are obtained with GPT-J on two-operand arithmetic queries.

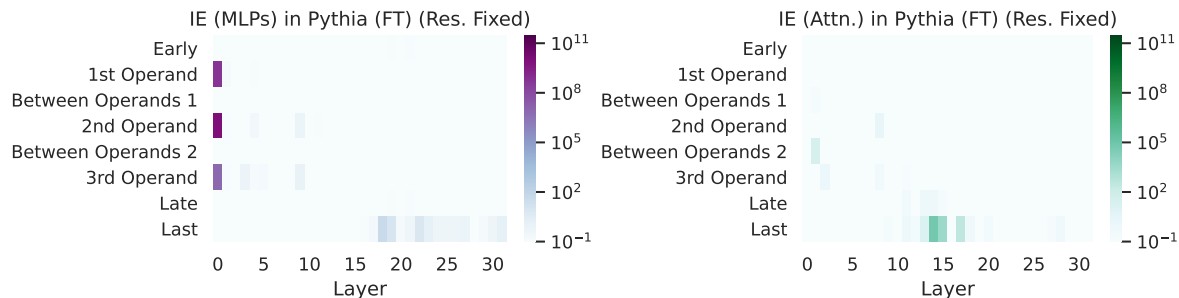

Figure 14: Indirect effect (IE) measured for the attention modules in the fine-tuned version of Pythia 2.8B on three-operand arithmetic queries, when $r = r'$.

## J   Additional Information Flow Visualizations

We include the IE measurements for the attention modules of GPT-J on the number retrieval task (Figure 11) and on the factual knowledge queries (Figure 12), and for Pythia 2.8B on three-operand arithmetic queries before and after fine-tuning (Figure 15). Additionally, we report the heatmap visualizations of the indirect effect measured for the following models: Pythia 2.8B (Figure 16), LLaMA 7B (Figure 17), Goat (Figure 18), and GPT-J using word numerals (Figure 19). Finally, we visualize in Figure 14 the IE of MLPs and attention modules for the fine-tuned Pythia 2.8B in the fixed-result case.

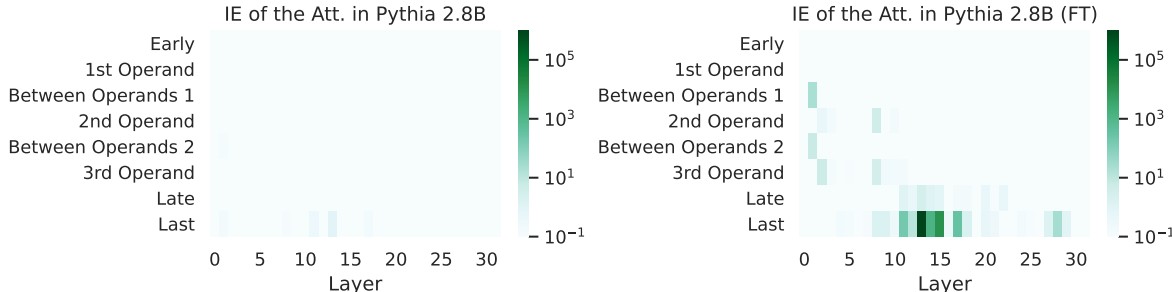

Figure 15: Indirect effect (IE) measured for the attention modules in Pythia 2.8B on three-operand arithmetic queries, before and after fine-tuning.

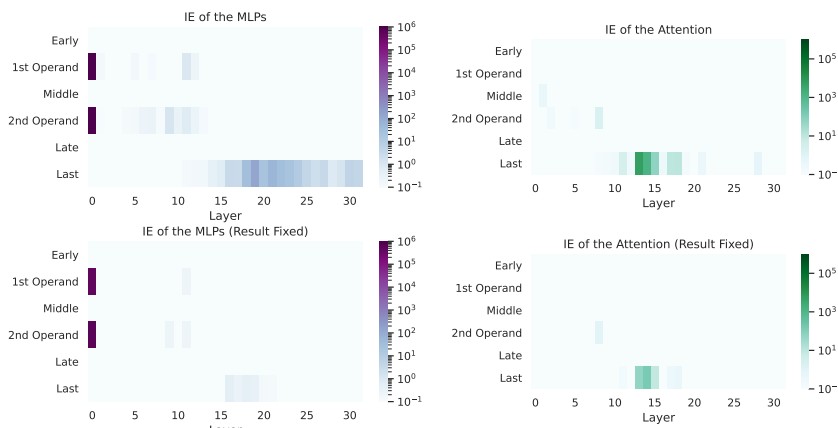

Figure 16: Indirect effect (IE) measured in the MLP and attention modules of Pythia 2.8B on two-operand arithmetic queries.

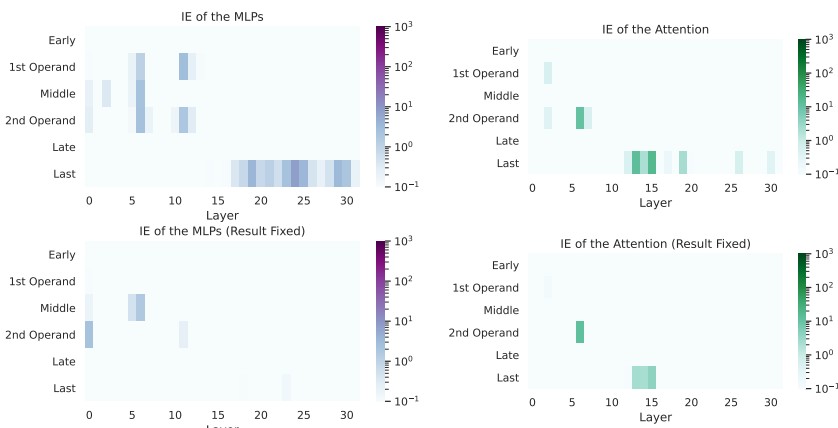

Figure 17: Indirect effect (IE) measured in the MLP and attention modules of LLaMA 7B on two-operand arithmetic queries.

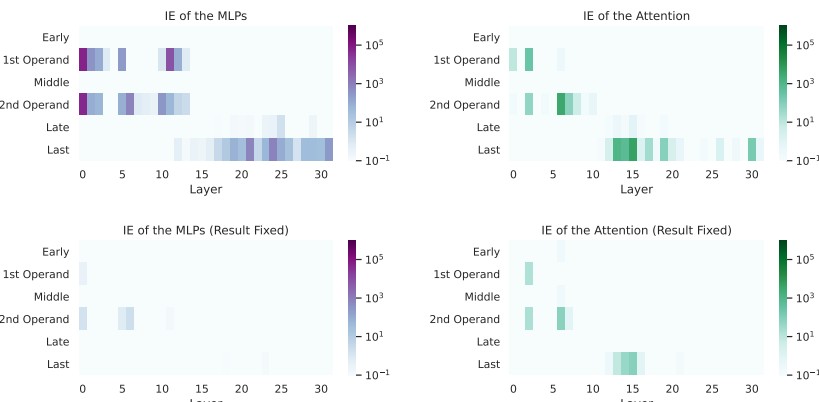

Figure 18: Indirect effect (IE) measured in the MLP and attention modules of Goat on two-operand arithmetic queries.

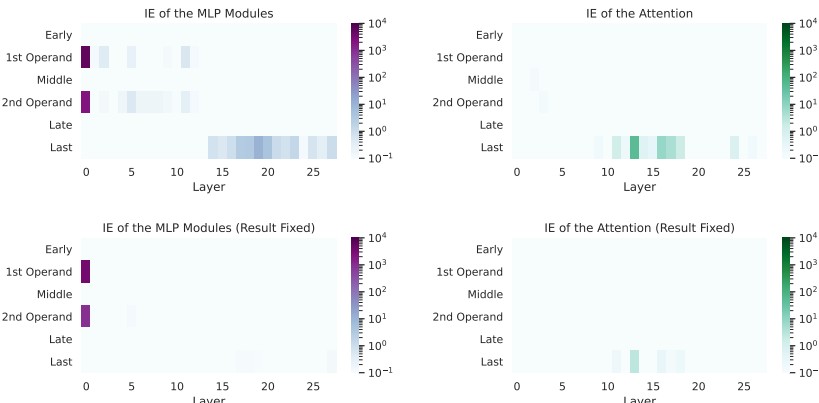

Figure 19: Indirect effect (IE) measured in the MLP and attention modules of GPT-J on two-operand arithmetic queries, using numeral words to represent quantities.