# OpenReview forum: "A Mechanistic Interpretation of Arithmetic Reasoning in Language Models using Causal Mediation Analysis"
_EMNLP/2023/Conference — EMNLP 2023 Main_

### Official Review · Reviewer_jY6n · 2023-08-03

**Typos Grammar Style And Presentation Improvements:** None
**Soundness:** 4

**Excitement:**

4: Strong: This paper deepens the understanding of some phenomenon or lowers the barriers to an existing research direction.

**Missing References:**

None

**Paper Topic And Main Contributions:**

The authors of this paper use causal analysis to locate components responsible for arithmetic reasoning in language models.

**Questions For The Authors:**

None

**Reasons To Accept:**

This paper provides interesting clues into how language models perform arithmetic reasoning. I find this to be an excellent application of causal analysis for NLP that both supports existing theories about how neural LMs work and provides a detailed picture of how the model components work together to produce an answer.  While there may not be an obvious immediate application for these findings, they are interesting enough to stand on their own. Furthermore, as Meng et al. (2022) show, causal mediation analysis (CMA) findings can yield useful applications further down the line.

**Reasons To Reject:**

My biggest concern is an issue that plagues CMA generally, which is over-interpreting results. This can come from assigning meaning to random patterns, or not properly considering counterfactual patterns, whether patterns are inevitable, and what exactly we can conclude from existing patterns. In my opinion, the authors do not sufficiently address some inevitabilities in the patterns. For instance, the model components associated with the "early" tokens will never have indirect effects because (as far as I can tell) there is no causal path from the intervention to the response variable that passes through these nodes. Similarly, the input MLPs for the "operand" tokens will always have high indirect effects because all causal paths must pass through these nodes. Lastly, there will always be a region in the components associated with the "last" token with high indirect effects whenever $r\neq r'$, since all causal paths must necessarily pass through some component in this layer. Given this a priori description of the effect patterns, it is hard to imagine the effect results looking much different from the way they do. To make this paper stronger, for each finding about the model's mechanism for arithmetic reasoning, the authors need to make a convincing case that there is an alternative hypothesis for the model's mechanism that would be supported by an alternative pattern that the authors can describe explicitly but does not show up in their experiments. The authors actually do an excellent job of this in section 4.2, but not so much in the other sections.

In conclusion, the authors should include an a priori description of what the patterns must look like in order to ground their actual results, and be explicit about what the alternatives to their findings would be. Including these would bring this paper to the next level in my opinion.

**Reproducibility:**

5: Could easily reproduce the results.

**Reviewer Confidence:**

4: Quite sure. I tried to check the important points carefully. It's unlikely, though conceivable, that I missed something that should affect my ratings.

---

> ### Author Rebuttal · Authors · 2023-08-28
>
> Thank you for finding our work to be “an excellent application of causal analysis for NLP” that provides “a detailed picture of how the model components work together to produce an answer”. We are encouraged by your statement “While there may not be an obvious immediate application for these findings, they are interesting enough to stand on their own”. Your review is greatly appreciated.
>
> > My biggest concern is an issue that plagues CMA generally, which is over-interpreting results.
>
> As the review correctly points out, some components are expected to have a large effect in mediating changes in specific parts of the input (e.g., the very first MLP layer in correspondence to the tokens that vary) and some components are expected to have no effect (e.g., modules that correspond to early tokens in the sequence). However, the statement “there will always be a region in the components associated with the ‘last’ token with high indirect effects whenever $r \neq r'$, since all causal paths must necessarily pass through some component in this layer” is not necessarily true. Late Transformer blocks at the end of the sequence are indeed likely to output information that has a high effect on the next token prediction (the last block at the last token is the only component through which all causal paths pass). However, this is the case for the output of the *whole* Transformer block (residual stream + attention output + MLP output). In our analyses, we are considering the output of MLPs and attention modules, which can be seen as "new information" being incorporated into the residual stream. This new information can be produced at any point of the sequence and then conveyed at the end of the sequence for the prediction of the next token.
>
> In other words, an alternative hypothesis for how the model processes the input to answer arithmetic queries is that information related to the result is computed by MLPs at different points during the input sequence (e.g., in correspondence to the second operand), and it is conveyed to the end of the sequence by the attention mechanism. For instance, for factual queries, we observed that the MLPs that have the largest effect are mid-sequence (corresponding to the subject tokens). In the discussion of these results, our goal was to point out that this effect pattern/information flow differs from the one observed for arithmetic queries, where the mid-late MLP layers have a larger influence on the model’s prediction.
> Our experiments show that, while some features of the effect patterns measured are predictable due to the nature of our experimental procedure (e.g., high effect in the first MLP layers corresponding to the modified tokens) the models display an information flow pattern that is specific to the task of answering arithmetic queries.
>
> In conclusion, we believe that your concern about the inevitability of some patterns in the effects measured is reasonable. We hope that our response clarifies what are other possible features in the information flow that we are measuring (i.e., what are possible alternative hypotheses), and we commit to making these considerations explicit and more clear in the discussion of all our results.

---

### Official Review · Reviewer_WmZT · 2023-08-03

**Soundness:** 3

**Excitement:**

4: Strong: This paper deepens the understanding of some phenomenon or lowers the barriers to an existing research direction.

**Paper Topic And Main Contributions:**

- The authors applied a ‘causal mediation analysis framework’ to the arithmetic reasoning of LMs and performed a mechanistic interpretation.
- The authors propose that they have discovered the mechanism by which LMs work in arithmetic tasks by intervening on specific model components, measuring changes in predicted probabilities, and discovering subsets of parameters related to specific predictions.

**Reasons To Accept:**

- As the authors mention, this study has novelty in attempting to connect the area of mechanistic interpretability with the mathematical reasoning ability of transformer-based LMs.

**Reasons To Reject:**

- Claiming that the discovery of changes in probability and differences in subsets of parameters is the discovery of the reasoning mechanism of LMs seems somewhat of a logical leap. The authors appear to implicitly assume that arithmetic reasoning in LM-based models operates like human-like reasoning. However, it is currently unclear whether LMs actually possess reasoning abilities. Therefore, there may be various interpretations as to what the changes that the authors have discovered actually mean.

- It is unclear what the ‘human-understandable component’ mentioned in the mechanistic interpretability refers to, and there is no argument presented as to how the causal mediation analysis framework proposed by the authors can be applied and appropriate to LMs. Therefore, it is difficult to verify the validity of the claim based on the given text alone.

- Also, the authors’ research is limited to a restricted level of arithmetic operations and does not cover the entire domain of arithmetic. In this situation, the claim that the mechanism of arithmetic reasoning has been discovered seems somewhat bold.

**Reproducibility:**

3: Could reproduce the results with some difficulty. The settings of parameters are underspecified or subjectively determined; the training/evaluation data are not widely available.

**Reviewer Confidence:**

3: Pretty sure, but there's a chance I missed something. Although I have a good feel for this area in general, I did not carefully check the paper's details, e.g., the math, experimental design, or novelty.

---

> ### Author Rebuttal · Authors · 2023-08-28
>
> Thank you for taking the time to review our work and for recognizing the novelty of our study.
>
> > **(1)** The authors appear to implicitly assume that arithmetic reasoning in LM-based models operates like human-like reasoning. However, **(2)** it is currently unclear whether LMs actually possess reasoning abilities.
>
> **Re. (1)**, we do not make any assumptions about any human-like behavior of the models. In fact, in the paper, we refrain from drawing any parallel between human reasoning and model mechanics: we analyze the information flow through the model, and indicate which components are responsible for different types of information. To avoid possible future misunderstandings, we will explicitly mention that we do not make such assumptions.
> **Re. (2)**, we are not making any claims about the reasoning capabilities of LMs. We are observing that the models we consider are performing the task of answering arithmetic queries with reasonable accuracy (60+%) and we are investigating how the models process the input information to perform such a task.
>
>
> > **(3)** It is unclear what the ‘human-understandable component’ mentioned in the mechanistic interpretability refers to, and **(4)** there is no argument presented as to how the causal mediation analysis framework proposed by the authors can be applied and appropriate to LMs.
>
> **Re. (3)**, our work studies the information flow within the model and the contribution of the different model components: we observe that the input information is conveyed by the attention to the end of the input sequence, where the information is processed by a specific subset of MLP modules, which output result-related information (Fig. 1). We believe that this procedure represents a human-understandable mechanism.
> **Re. (4)**, approaches based on causal mediation analysis have been applied and shown to be appropriate to interpret language models in multiple previous works [1, 2, 3, 4]. Our framework relies on the principle that, by intervening on a particular mediator (a set of neurons) in a causal graph (a model), we can isolate the causal influence of that mediator on the outcome variable (the model’s prediction). Building our analyses on this logically-sound procedure, we study the causal influence of different model components.
>
>
> > The authors’ research is limited to a restricted level of arithmetic operations and does not cover the entire domain of arithmetic.
>
> Arithmetic consists of the study of the properties of the traditional operations on numbers—addition, subtraction, multiplication, division, exponentiation, and extraction of roots. Among these, we experiment with the four fundamental binary operators. Addition, subtraction, multiplication, and division form the cornerstone of arithmetic calculations and serve as the basis for a wide range of mathematical computations. Therefore, exploring their mechanisms in language models provides a starting point to explore more complex forms of mathematical processing. As an initial study in this direction, our work represents the groundwork that can inform subsequent investigations into a broader set of operations.
>
> [1] Finlayson, M., et al., 2021, August. Causal Analysis of Syntactic Agreement Mechanisms in Neural Language Models. In Proceedings of the 59th Annual Meeting of the Association for Computational Linguistics and the 11th International Joint Conference on Natural Language Processing (Volume 1: Long Papers).
> [2] Meng, K., et al., 2022. Locating and editing factual associations in GPT. Advances in Neural Information Processing Systems, 35, pp.17359-17372.
> [3] Vig, J., et al., 2020. Investigating gender bias in language models using causal mediation analysis. Advances in neural information processing systems, 33, pp.12388-12401.
> [4] Wang, K.R., et al., 2022, September. Interpretability in the Wild: a Circuit for Indirect Object Identification in GPT-2 Small. In The Eleventh International Conference on Learning Representations.

---

### Official Review · Reviewer_LE67 · 2023-08-08

**Soundness:** 4

**Excitement:**

4: Strong: This paper deepens the understanding of some phenomenon or lowers the barriers to an existing research direction.

**Paper Topic And Main Contributions:**

The paper uses causal mediation analysis to examine how Transformer-based LMs process arithmetic tasks. By intervening on specific model parts, they found that LMs transfer arithmetic-relevant information from mid-sequence early layers to the final token, which is then processed by late MLP modules. They supported this finding by experimenting on autoregressive LMs in different scales and highlighting their information flow's specificity to arithmetic tasks.

**Reasons To Accept:**

* The paper is well-written, very clear, and concise.
* Provides empirical evidence supporting the proposed hypothesis by conducting an extensive evaluation on LMs in 4 different scales.
* Offer interesting insights into how arithmetic reasoning is conducted in Transformer based LMs.

**Reasons To Reject:**

* Limited Scope: The paper focuses on a small set of arithmetic operations.

**Reproducibility:**

5: Could easily reproduce the results.

**Reviewer Confidence:**

3: Pretty sure, but there's a chance I missed something. Although I have a good feel for this area in general, I did not carefully check the paper's details, e.g., the math, experimental design, or novelty.

**Typos Grammar Style And Presentation Improvements:**

* It would be helpful to add additional details in the intro. For example lines 77-80 and 86-90 (what are the additional tasks).
* Fig 2. The "before and after intervention" plots are too small to read. (same for Fig 3)

---

> ### Author Rebuttal · Authors · 2023-08-28
>
> Thank you for recognizing that our work offers “interesting insights” into arithmetic reasoning in LMs obtained “by conducting an extensive evaluation”. Also, thank you for your presentation suggestions, we will incorporate them in the final version of the paper.
>
> > The paper focuses on a small set of arithmetic operations.
>
> The scope of our paper is investigating arithmetic reasoning and we experiment with the four fundamental arithmetic operators. Addition, subtraction, multiplication, and division form the cornerstone of arithmetic calculations and serve as the basis for a wide range of mathematical computations. Thus, exploring their mechanisms in language models provides a starting point to explore more complex forms of mathematical processing. As a first study in this direction, our work lays the foundation that can inform subsequent investigations into a broader set of operations.

---

### Official Review · Reviewer_yvrj · 2023-08-11

**Soundness:** 4

**Excitement:**

3: Ambivalent: It has merits (e.g., it reports state-of-the-art results, the idea is nice), but there are key weaknesses (e.g., it describes incremental work), and it can significantly benefit from another round of revision. However, I won't object to accepting it if my co-reviewers champion it.

**Paper Topic And Main Contributions:**

The paper addresses the topic of mathematical reasoning in large language models, specifically how Transformer-based models process arithmetic tasks. Using a causal mediation analysis framework, the research identifies key model components responsible for arithmetic predictions, revealing that information is transmitted from mid-sequence layers to the final token via attention mechanisms and then processed by MLP modules before being integrated into the residual stream.

**Reasons To Accept:**

Its strengths lie in its mechanistic interpretation, utilizing a causal mediation analysis framework to understand LMs' inner workings. By intervening on specific model activations, the study effectively traces information flow and pinpoints key components, such as attention mechanisms and MLP modules, that influence arithmetic predictions. For the NLP community, this paper promises an enhanced understanding of LMs' mathematical capabilities. The presentation of the paper has good standards.

**Reasons To Reject:**

While the paper provides a deep dive into how LMs handle mathematical reasoning, it stops short of offering tangible methods to adjust or improve the LMs based on the gleaned insights. This leaves a gap between understanding and application. Interpretability, while crucial, is just one aspect of model research. By intensely focusing on it, the study might not address other pressing issues related to model efficiency, robustness, or scalability.

**Reproducibility:**

3: Could reproduce the results with some difficulty. The settings of parameters are underspecified or subjectively determined; the training/evaluation data are not widely available.

**Reviewer Confidence:**

3: Pretty sure, but there's a chance I missed something. Although I have a good feel for this area in general, I did not carefully check the paper's details, e.g., the math, experimental design, or novelty.

---

> ### Author Rebuttal · Authors · 2023-08-28
>
> Thank you for pointing out that our study “effectively traces information flow and pinpoints key components” involved in arithmetic reasoning in LMs. Also, thank you for recognizing that our work “promises an enhanced understanding of LMs' mathematical capabilities” and is relevant to the NLP community.
>
> > Interpretability, while crucial, is just one aspect of model research.
>
> The review motivates the low score with the point that interpretability research is not valuable without any proposal for model improvement.
> We believe that it is important to acknowledge that understanding the inner workings of models is a crucial step toward informed decision-making in any model-related endeavor. Interpretability serves as a vital tool for identifying potential strengths, limitations, and areas for enhancement in LMs. As pointed out by another reviewer, similar interpretability studies (Meng et al., 2022; Nanda et al., 2023) yield useful applications further down the line.
>
> Moreover, the review assigns a reproducibility score of 2. We find ourselves a little perplexed as we provided, through the submission platform, the code with the instructions to reproduce all our experiments, as well as notebooks to generate the visualizations included in the paper. We would greatly appreciate  revision of such a score, or otherwise a clarification about any missing details for reproduction.
>
>
> [1] Nanda, N., et al., 2022, September. Progress measures for grokking via mechanistic interpretability. In The Eleventh International Conference on Learning Representations.
> [2] Meng, K., et al., 2022. Locating and editing factual associations in GPT. Advances in Neural Information Processing Systems, 35, pp.17359-17372.

---

### Meta-Review · Area_Chair_X5qY · 2023-09-11

**Recommendation:** 4

**Metareview:**

This paper proposes a mechanistic interpretation of Transformer-based LMs on arithmetic-based questions using a causal mediation analysis. They identify the subset of parameters responsible for specific predictions.

The main contribution of the paper is the causal mediation analysis (CMA) on arithmetic.

In pros, reviewers praise that the paper is well written, provides a better understanding of the mathematical capabilities of transformers / interesting insights, with findings that are "interesting enough to stand on their own".

The main cons / weakness is the limited applicability of an analysis only performed on arithmetic / the limited scope, and that this does not lead to any insights to improve transformers with. There are a few worries on that the claims are a little too bold, which could be adjusted by the authors should the paper be accepted.

All in all, the reviewers were (almost) unanimous in that the paper is strong in soundness (3/4) and strong in excitement (3/4), and the only (weak) reason to reject it would be its limited scope to arithmetic.

---

### Decision · Program_Chairs · 2023-10-07

**Decision:**

Accept-Main

**Comment:**

This paper proposes a mechanistic interpretation of Transformer-based LMs on arithmetic-based questions using a causal mediation analysis. They identify the subset of parameters responsible for specific predictions.

The main contribution of the paper is the causal mediation analysis (CMA) on arithmetic.

In pros, reviewers praise that the paper is well written, provides a better understanding of the mathematical capabilities of transformers / interesting insights, with findings that are "interesting enough to stand on their own".

The main cons / weakness is the limited applicability of an analysis only performed on arithmetic / the limited scope, and that this does not lead to any insights to improve transformers with. There are a few worries on that the claims are a little too bold, which could be adjusted by the authors should the paper be accepted.

All in all, the reviewers were (almost) unanimous in that the paper is strong in soundness (3/4) and strong in excitement (3/4), and the only (weak) reason to reject it would be its limited scope to arithmetic.